# Cardiometabolic Health in Relation to Lifestyle and Body Weight Changes 3–8 Years Earlier

**DOI:** 10.3390/nu10121953

**Published:** 2018-12-10

**Authors:** Tessa M. Van Elten, Mireille. N. M. Van Poppel, Reinoud J. B. J. Gemke, Henk Groen, Annemieke Hoek, Ben W. Mol, Tessa J. Roseboom

**Affiliations:** 1Department of Public and Occupational Health, Amsterdam UMC, Vrije Universiteit Amsterdam, De Boelelaan, 1117 Amsterdam, The Netherlands; mireille.van-poppel@uni-graz.at (M.N.M.V.P.); rjbj.gemke@vumc.nl (R.J.B.J.G.); 2Department of Clinical Epidemiology, Biostatistics and Bioinformatics, University of Amsterdam, Amsterdam UMC, Meibergdreef 9, 1105 AZ Amsterdam, The Netherlands; t.j.roseboom@amc.uva.nl; 3Department of Obstetrics and Gynecology, University of Amsterdam, Amsterdam UMC, Meibergdreef 9, 1105 AZ Amsterdam, The Netherlands; 4Amsterdam Public Health Research Institute, 1105 AZ Amsterdam, The Netherlands; 5Amsterdam Reproduction and Development, 1105 AZ Amsterdam, The Netherlands; 6Institute of Sport Science, University of Graz, 8010 Graz, Austria; 7Department of Pediatrics, Emma Childrens Hospital, Vrije Universiteit Amsterdam, Amsterdam UMC, 1081 HV Amsterdam, The Netherlands; 8Department of Epidemiology, University of Groningen, University Medical Centre Groningen, 9700 RB Groningen, The Netherlands; h.groen01@umcg.nl; 9Department of Obstetrics and Gynecology, University of Groningen, University Medical Centre, Groningen, 9700 RB Groningen, The Netherlands; a.hoek@umcg.nl; 10School of Medicine, The Robinson Institute, University of Adelaide, 5006 Adelaide, Australia; b.w.mol@amc.uva.nl; 11Department of Obstetrics and Gynecology, Monash University, 3800 Melbourne, Australia

**Keywords:** dietary intake, physical activity, body weight, lifestyle change, cardiometabolic health, long-term follow-up

## Abstract

The degree to which individuals change their lifestyle in response to interventions differs and this variation could affect cardiometabolic health. We examined if changes in dietary intake, physical activity and weight of obese infertile women during the first six months of the LIFEstyle trial were associated with cardiometabolic health 3–8 years later (*N* = 50–78). Lifestyle was assessed using questionnaires and weight was measured at baseline, 3 and 6 months after randomization. BMI, blood pressure, body composition, pulse wave velocity, glycemic parameters and lipid profile were assessed 3–8 years after randomization. Decreases in savory and sweet snack intake were associated with lower HOMA-IR 3–8 years later, but these associations disappeared after adjustment for current lifestyle. No other associations between changes in lifestyle or body weight during the first six months after randomization with cardiovascular health 3–8 years later were observed. In conclusion, reductions in snack intake were associated with reduced insulin resistance 3–8 years later, but adjustment for current lifestyle reduced these associations. This indicates that changing lifestyle is an important first step, but maintaining this change is needed for improving cardiometabolic health in the long-term.

## 1. Introduction

Unhealthy diet, physical inactivity and a BMI over 25 kg/m^2^ are well known risk factors for cardiovascular diseases (CVDs). It is therefore that primary prevention of CVDs focuses, among others, on improving these lifestyle factors and reducing body weight [1]. Observational studies showed graded relationships between healthy changes in dietary intake, physical activity and body weight over time and future cardiometabolic health. For example, improvements in diet [2], or transition to a more active lifestyle [3] lowered risk of coronary heart disease among women in the Nurses’ Health Study.

Randomized controlled trials (RCTs) showed that improving lifestyle and reducing body weight improved cardiometabolic health [4,5]. However, although lifestyle interventions can improve cardiometabolic health, the effectiveness of lifestyle interventions varies [6,7]. For example, a meta-analysis including 54 RCTs to determine the effect of aerobic exercise on blood pressure showed that net change in blood pressure after following a physical activity intervention varied from −16.7 to 3.9 mmHg for systolic blood pressure and from −11.0 to 11.3 mmHg for diastolic blood pressure [8]. Despite the large variation in lifestyle and body weight change, the effects of a lifestyle intervention are often described as a randomized group comparison. By studying the effect of absolute change in lifestyle and body weight, more knowledge can be gathered on dose-response relationships and potential thresholds between lifestyle and body weight change with later life cardiometabolic health. This information will aid in improving primary prevention.

The LIFEstyle RCT enrolled obese infertile women, and allocated them to a six-month diet and physical activity intervention or to prompt infertility treatment [9,10,11]. We previously showed that the intervention lowered the intake of high caloric snacks and beverages and increased physical activity in the short term [12], reduced body weight [11] and improved cardiometabolic health at the end of the six-month intervention period [13]. Furthermore, women allocated to the intervention group reported a lower energy intake at 5.5 years after randomization [14]. Individual responses to the lifestyle intervention varied largely among study participants. The mean weight change in the intervention group was −4.4 kg and the standard deviation of 5.8 kg underlines this large variation [11]. Also women allocated to the control group changed their lifestyle: 10.5% of the women in the control group lost 5% or more of their original body weight [11]. Hence we here investigate individual changes in lifestyle and body weight and relate these changes in dietary intake, physical activity and weight during the first six months after randomization to cardiometabolic health 3–8 years later.

We hypothesized that women who increased their intake of vegetables and fruit, decreased their intake of sugary drinks and snacks, became more physically active and lost more weight during the first six months after randomization had a better cardiometabolic health 3–8 years after randomization compared to women who did not show these improvements in lifestyle and weight. We therefore examined if the change in dietary intake, physical activity and body weight of obese infertile women, combining the intervention and control group, over the first six months of a preconception lifestyle intervention study was associated with their cardiometabolic health 3–8 years after the start of the study.

## 2. Materials and Methods

### 2.1. Study Population

The study population comprises all women who participated in the follow-up of the LIFEstyle study. The LIFEstyle study was a multicenter randomized controlled trial (RCT), conducted between 2009 and 2014 in the Netherlands [9,10,11]. In total, 577 women between 18 and 39 years old, with a BMI of ≥29 kg/m^2^ were randomized into a six-month structured lifestyle intervention (intervention group) or infertility care as usual (control group). The lifestyle intervention focused on eating a healthy diet according to the Dutch Dietary Guidelines 2006 [15], including a caloric reduction of 600 kcal/day but not below 1200 kcal/day, and being physically active 2–3 times a week for at least 30 min at moderate intensity (60–85% of maximum heart rate frequency). Women were additionally advised to increase physical activity in daily life by taking at least 10,000 steps per day.

At 3–8 years after randomization, 574 women were eligible to participate in the follow-up study of the LIFEstyle RCT [16]. During the follow-up study, data were collected using questionnaires and physical examinations. In the current study, we included all women that participated in the physical examinations (*N* = 111; Figure 1).

The study was conducted in accordance with the Declaration of Helsinki and all procedures were approved by the Medical Ethics Committee of the University Medical Centre Groningen, the Netherlands (METc 2008/284). Written informed consent was obtained from all participants at the start of the LIFEstyle study as well as the start of the follow-up.

### 2.2. Assessment of Dietary Intake, Physical Activity and Body Weight to Calculate Change

In order to calculate change in lifestyle and body weight over the first six months of the study, we used dietary intake, physical activity and body weight measures collected at baseline, 3 and 6 months after randomization in the lifestyle intervention study. Dietary intake was examined using a 33-item food frequency questionnaire (FFQ) consisting of two parts. The first part was based on the standardized questionnaire on food consumption used for the Public Health Monitor in the Netherlands [17], asking about the type of cooking fats used, the consumed type of bread, frequency of breakfast use, frequency of consumption and portion size of vegetables, fruits and fruit juice. The questions on the intake of fruit, fruit juice and cooked vegetables were validated against two 24-h recalls. The estimated intake of fruit and fruit juice consumption based on the questionnaire showed fairly strong comparability with the intake based on the two 24-h recalls, however the comparability for cooked vegetables was weak [17]. The second part consisted of additional questions about savory and sweet snack intake and the intake of soda. For all foods, frequency of consumption per week or per months was asked and portion size was asked per standard household measure. The presented portion sizes and food groups in the current study were pre-specified in the questions of the FFQ. Dietary intake was studied as the intake of vegetables (raw as well as cooked; grams/day), fruits (grams/day), sugar containing beverages (fruit juice and soda; glasses/day), savory snacks (crisps, pretzels, nuts and peanuts; handful/week) and sweet snacks (biscuits, pieces of chocolate, candies or liquorices; portion/week). One portion of sweet snacks included 2 biscuits, or 2 pieces of chocolate, or 5 candies, or 5 pieces of liquorice.

Physical activity was examined using the validated Short QUestionnaire to ASsess Health-enhancing physical activity (SQUASH) questionnaire [18]. This questionnaire asked about the number of days per week, the average time per day or week (hours and/or minutes), and the intensity (low, moderate, high) of physical activity in four domains: commuting activities, leisure time activities, household activities, and activities at work and school. Physical activity was studied as total moderate to vigorous physical activity (MVPA; hours/week).

Body weight (kg) was measured during hospital visits at baseline, 3 months and 6 months after randomization by trained research nurses that were not involved in the lifestyle intervention coaching.

### 2.3. Cardiometabolic Health at Follow-Up

Cardiometabolic health 3–8 years after randomization was examined in a mobile research vehicle by two researchers, using a standardized research protocol. Women were asked not to eat or drink from 90 min onwards before the mobile research vehicle arrived at their home. They were additionally asked not to drink caffeine containing beverages or to smoke from 12 h onwards before the physical examination. Height and current weight were measured to calculate current BMI. Height was measured to the nearest 0.1 cm using a wall stadiometer (SECA 206; SECA, Germany) on bare feet, with heels flat on the ground at an angle of 90 degrees, head in Frankfort horizontal position and heels, back and shoulders straight against the wall. Current weight was measured in underwear to the nearest 0.1 kg using a digital weighting scale (SECA 877; SECA, Germany), while the participant was standing still and looking straight ahead. All measurements were done twice, and in case of >0.5 cm difference in height and >0.5 kg difference in weight, a third measurement was performed. After sitting quietly for 5 min, blood pressure was measured three times at heart level, at the non-dominant arm, using an automatic measurement device (Omron HBP-1300; OMRON Healthcare, The Netherlands) with appropriate cuff size. The three measurements were done using a 30 s time interval and women were not allowed to talk in between, move, cross their legs or tense their arm muscles. Body composition was measured twice by bio-electrical impedance (BIA; Bodystat 1500; Bodystat Ltd., Isle of Man, UK) after lying quietly for 5 min. On beforehand participants were asked to take of any jewelry, belts, piercings, etc. which could affect the BIA measurements. After cleaning the skin with alcohol, electrode strips were attached at the dorsal side of the left hand and foot with at least 3–5 cm in between the two electrodes. Women were instructed not to talk in between the measurements and attention was paid that arms and legs did not touch other parts of the body. A third measurement was performed in case the impedance or resistance differed >5Ω. Fat mass and fat free mass were calculated using equation of Kyle and colleagues [19]. Immediately after the BIA measurement, still in supine position, carotid-femoral pulse wave velocity (PWV) was measured twice using the Complior Analyse (Complior; Alam Medical, Saint-Quentin-Fallavier, France). Mechanotransducer censors were placed at the carotid artery on the right side and on the femoral artery on the left side. Blood pressure in lying position was measured once before the actual measurement started and entered in the Complior software. Directly after the measurement, distance between both censors was measured and also entered in the Complior software. In case of >10% difference in PWV between both measurements, a third measurement was done. The following equation was used to calculate PWV: PWV = 0.8 × (distance between the carotis and fermoralis measuring site/Δ time between upstrokes of pressure waves) [20].

Apart from the physical examinations, a trained nurse visited the participants at home to draw a venous blood sample after an overnight fast. All venous blood samples were analyzed at the biochemical laboratory of the Amsterdam UMC. We examined metabolic health by fasting serum concentrations of glucose (Roche cobas 8000, c702; Roche Diagnostics, Rotkreuz, Switzerland) and insulin (Centaur XP; Siemens, Munich, Germany), triglycerides (Roche cobas 8000, c702; Roche Diagnostics, Switzerland), total cholesterol (Roche cobas 8000, c502; Roche Diagnostics, Switzerland) and high density (HDL-C) lipoprotein cholesterol (Roche cobas 8000, c702; Roche Diagnostics, Switzerland). Low density (LDL-C) lipoprotein cholesterol was calculated using the following formula; (total cholesterol) − (high density lipoprotein cholesterol) − 0.45 × (triglycerides). Furthermore, insulin resistance (HOMA-IR) was calculated as fasting insulin concentration in μU/mL multiplied by fasting glucose concentration in mmol/L divided by 22.5. We additionally examined if metabolic syndrome was present or not. If present, participant met at least three of the following criteria determined by the American Heart Association: glucose ≥ 5.6 mmol/L, HDL-C < 1.3 mmol/L, triglycerides ≥ 1.7 mmol/L, waist circumference ≥ 88 cm or blood pressure ≥ 130/85 mmHg [21].

### 2.4. Statistical Analysis

We used linear regression models to study the association between the change in dietary intake, physical activity, and weight during the first six months after randomization and cardiometabolic health 3–8 years later. Results are displayed as betas (β) and 95% confidence intervals (C.I.). For metabolic syndrome, logistic regression was used and results are displayed as odds ratios (OR) and 95% C.I. We recalculated the intake of vegetables and fruits into 10 g/day dividing the change by ten. Our regression models therefore display the effect on cardiometabolic health per 10 g change of vegetable intake and fruit intake per day.

For the descriptive statistics the change in lifestyle and body weight over time was calculated by subtracting the baseline measurement from the last know measurement at preferably 6 months or otherwise 3 months after randomization (Appendix A). In most women, the last known measurement was at six months after randomization, but when missing, the measurement at three months after randomization was used (*N* = 5). This means that a higher change score for vegetable intake, fruit intake, and MVPA is healthier, while a higher change score for sugary drink intake, savory and sweet snack intake, and weight change (higher means weight gain instead of weight loss) is unhealthier. In our regression models, the change in vegetable intake, fruit intake and MVPA was calculated by subtracting the baseline measurement from the last known measurement. The change in sugary drinks, savory and sweet snack intake and body weight was calculated by subtracting the last known measurement from the baseline measurement. An increase in change score in our regression models is favorable, reflecting an increase in vegetable intake, fruit intake and MVPA, and a decrease in sugary drink intake, savory and sweet snack intake and body weight.

All crude regression models were corrected for baseline cardiometabolic health, depending on the outcome variable (e.g., BMI at follow-up was corrected for BMI at baseline), with exception of fat mass, fat free mass, and PWV as these outcomes were not measured at baseline. We therefore corrected fat mass and fat free mass models for baseline BMI [22], and PWV models for baseline systolic blood pressure [23]. Based on literature [24] and the associations with both the change variables and the outcome variables, we corrected the adjusted regression models additionally for (1) smoking at follow-up (yes/no; self-reported by means of a questionnaire); (2) current dietary intake, depending on which food group is added into the model; e.g., if we examined the association between the change in vegetable intake during the first six months after randomization and current BMI, we corrected for current vegetable intake, and when examining the effect of MVPA we corrected for total current energy intake (kcal/day). Current dietary intake, i.e., dietary intake during follow-up, was assessed with the same 33-item FFQ as described previously [17]; (3) current MVPA (min/day), i.e., MVPA during follow-up, which was measured for seven consecutive days using the triaxial Actigraph wGT3X-BT or GT3X+ accelerometer [25]. Freedson cut-off points were used to determine the number of minutes per day in MVPA (≥1952 counts/min) [26]. We additionally added randomization group (intervention/control group) and time between randomization and follow-up (years) into the adjusted model to see if this affected the effect estimates. As women got pregnant during the LIFEstyle study, some diet, physical activity and weight measurements were collected during early pregnancy. We therefore once excluded measurements collected during pregnancy to see if effect estimates changed.

Statistical analyses were performed using the software Statistical Package for the Social Sciences (SPSS) version 24 for Windows (SPSS, Chicago, IL, USA). *p*-values <0.05 were considered statistically significant.

## 3. Results

Of the 577 women randomly allocated to the intervention and control group during the trial, 574 women were eligible to participate in the physical examinations at 3–8 years after the intervention (Figure 1). Of these eligible women, 121 were willing to participate in the follow up study and signed informed consent (21.1%) and we collected data of 111 women. Because of missing data regarding the change in dietary intake, physical activity, weight and the covariates, we were able to include 50 up to 78 women in our regression analyses.

Mean age of the women during physical examination was 36.4 years (SD = 4.3), most of them were Caucasian (94.6%), had an intermediate vocational education (49.1%) and were obese at 3–8 years after the intervention (mean BMI = 35.5 kg/m2 (SD = 5.3); Table 1). Baseline characteristics, collected during the LIFEstyle RCT, did not differ between participants (*N* = 111) and non-participants (*N* = 463) of the follow-up study (*N* = 111), with exception of ethnicity and the change in sweet snack intake (Appendix A). Participants were more often of Caucasian origin (94.6%) compared to non-participants (85.7%). Additionally, the change in sweet snacks during the first six months after randomization was lower in the participants (−0.1 portions/week (SD = 5.6)) compared to the non-participants (−3.3 portions/week (SD = 10.2)).

Table 2 shows the change in dietary intake, physical activity and body weight during the first six months of the LIFEstyle study in our study population.

We did not observe the hypothesized associations between increases in vegetable intake, fruit intake and MVPA during the first six months of the LIFEstyle study with a more favorable BMI, blood pressure, body composition, PWV and metabolic health 3–8 years later (Table 3). Furthermore, decreased sugary drink intake and body weight were not associated with a more favorable cardiovascular health 3–8 years later. A decrease in savory snacks intake during the first six months after randomization was associated with a lower HOMA-IR at follow-up (crude model: −0.16 (−0.32; −0.001); *p* = 0.049). This association disappeared after adjustment for smoking, current savory snack intake and current MVPA (adjusted model: −0.09 (−0.28; 0.09); *p* = 0.33). Changes in savory snack intake were not associated with women’s BMI, blood pressure, body composition, PWV or other metabolic outcomes at follow-up. Furthermore, a decrease in sweet snacks intake during the first six months after randomization was associated with a lower HOMA-IR at follow-up (crude model: −0.16 (−0.06; −0.06); *p* = 0.003). Also this association disappeared after adjustment for smoking, current sweet snack intake and current MVPA (adjusted model: 0.01 (−0.09; 0.12); *p* = 0.84).

Dietary intake, physical activity and weight change during the first six months after randomization were not associated with having metabolic syndrome at follow-up (Table 4).

Adding randomization arm into the model (intervention or control group) and time between randomization and follow-up (years) did not change effect estimates. Additionally, effect estimates hardly changed when we excluded women in early pregnancy from our study sample (results not shown).

## 4. Discussion

A decrease in savory and sweet snacks during the first six months after randomization was associated with lower insulin resistance 3–8 years later. However, these associations became non-significant after adjustment for current lifestyle. No other associations between changes in lifestyle or body weight during the first six months after randomization with cardiovascular health 3–8 years later were observed.

A reason why we did not observe statistically significant associations between changes in lifestyle and body weight during the first six months after randomization with cardiovascular health 3–8 years later might be the low number of participants included in the follow-up study, and therefore we had low power to observe the hypothesized associations. Furthermore, it might be that there was not enough individual variation in the changes in lifestyle and body weight to find any associations with cardiovascular health at follow-up, which could be explained by the fact that women allocated to the control group participated to a larger extent in our follow-up study than women allocated to the intervention group.

There were multiple associations pointing towards our hypothesis that healthy changes in lifestyle are associated with more favorable cardiovascular health. Our findings that women with a higher intake of fruit (*p* = 0.06) and more MVPA (*p* = 0.09) had a higher fat free mass are in accordance with physiological mechanisms. Muscular activity stimulates the development and maintenance of lean muscle mass [27] and the high fiber content of the diet, associated with fruit intake, is related to lower body fat [28]. Furthermore, women who lost weight during the first six months after randomization tended to have higher HDL-cholesterol (*p* = 0.06). This is in line with findings in other studies and may relate to links between lower visceral fat mass and higher HDL-cholesterol [29].

However, we also observed that women who decreased their sugary drink intake tended to have a lower HDL-cholesterol (*p* = 0.06), which is not in line with our hypothesis and unexpected, assuming that a lower intake of sugary drinks causes a lower BMI, which is associated with improved HDL-cholesterol [29]. We do not know why we observed this: additional corrections for current BMI did not weaken this association and high sugary drink intake was not correlated with high MVPA levels, which is associated with higher HDL-cholesterol [30]. We were not able to analyze the effect of fresh fruit juice on HDL-cholesterol, because the 33-item FFQ do not ask specifically about fresh fruit juice. Evidence showed that fresh fruit juice might be associated with lower HDL-cholesterol [31]. It could therefore be that women specifically reduced their intake of sugar sweetened fruit juice, but not fresh fruit juice, which might have led to the unexpected association between decreases in sugary drinks and lower HDL-cholesterol. However, given the number of associations studied, it might also be that this is a chance finding.

Our results of reduced snack intake and improved HOMA-IR indicate that changing lifestyle is an important first step, but that maintaining a healthy lifestyle is needed for improving cardiometabolic health in the long-term. However, lifestyle change and maintaining those healthy changes on the long-term is notoriously difficult. To sustain long-term intervention adherence and thereby improve cardiometabolic health, it might be helpful to provide extended care by offering long-term individual or group contact to stimulate healthy behavior [32].

An important strength of the current study is the detailed information about the intake of specific foods and beverages, physical activity and body weight during the first six months after randomization into a preconception lifestyle program. This enabled us to gain more knowledge about lifestyle and body weight changes during the first six months after randomization in association with cardiometabolic health 3–8 years after the intervention instead of a randomized comparison between groups. We additionally had good quality data (measured by trained researchers) about cardiometabolic health at the start of the intervention and at 3–8 years after randomization, and were able to take into account women’s baseline cardiometabolic health. There are also limitations that should be mentioned. Dietary intake as well as physical activity was measured using self-reported questionnaires. Obese women tend to under-report unhealthy behavior and over-report healthy behavior [33], and women allocated to the intervention group might do this to a larger extent because of social desirability bias [34,35]. However, adding randomization group into our regression models hardly changed the effect estimates, which indicates that the effect of social desirability bias induced by randomization group is minimal. Furthermore, there was a wide range (3–8 years) in the time between inclusion in the preconception lifestyle intervention study and our follow-up assessment of cardiometabolic health. This wide range might have affected the associations between lifestyle and body weight change with cardiometabolic health at follow-up. However, adding time in between randomization and follow-up into our models hardly changed the effect estimates. Finally, the 33-item FFQ pre-specified two food groups, fruit juice and savory snack intake, including foods known to have favorable as well as unfavorable effects on cardiometabolic health. The question on consumption of fruit juice does not distinguish between fresh fruit juice and sugar sweetened juice, while studies show that the consumption of fresh juice reduces cardiovascular risk factors due to, amongst others, the antioxidant effects and anti-inflammatory effects [31]. Furthermore, the question on savory snack consumption combines the intake of crisps, pretzels, nuts and peanuts into one question, while studies show that nuts and peanuts might be beneficial for cardiovascular health [36]. It might therefore be that an increase in these food groups represents a healthy change instead of an unhealthy change. We recommend future research to use a more extensive FFQ, not pre-specifying these foods into one food group. Future research should replicate our results in a larger study population, preferably with larger variations in lifestyle and weight changes.

## 5. Conclusions

To conclude, decreases in savory and sweet snack intake were associated with reduced insulin resistance 3–8 years later, but after adjustment for current lifestyle these associations disappeared. No other associations between changes in lifestyle or body weight during the first six months after randomization with cardiovascular health 3–8 years later were observed. Changing lifestyle is an important first step, but maintaining this change is needed to improve cardiometabolic health in the long-term.

## Figures and Tables

**Figure 1 nutrients-10-01953-f001:**
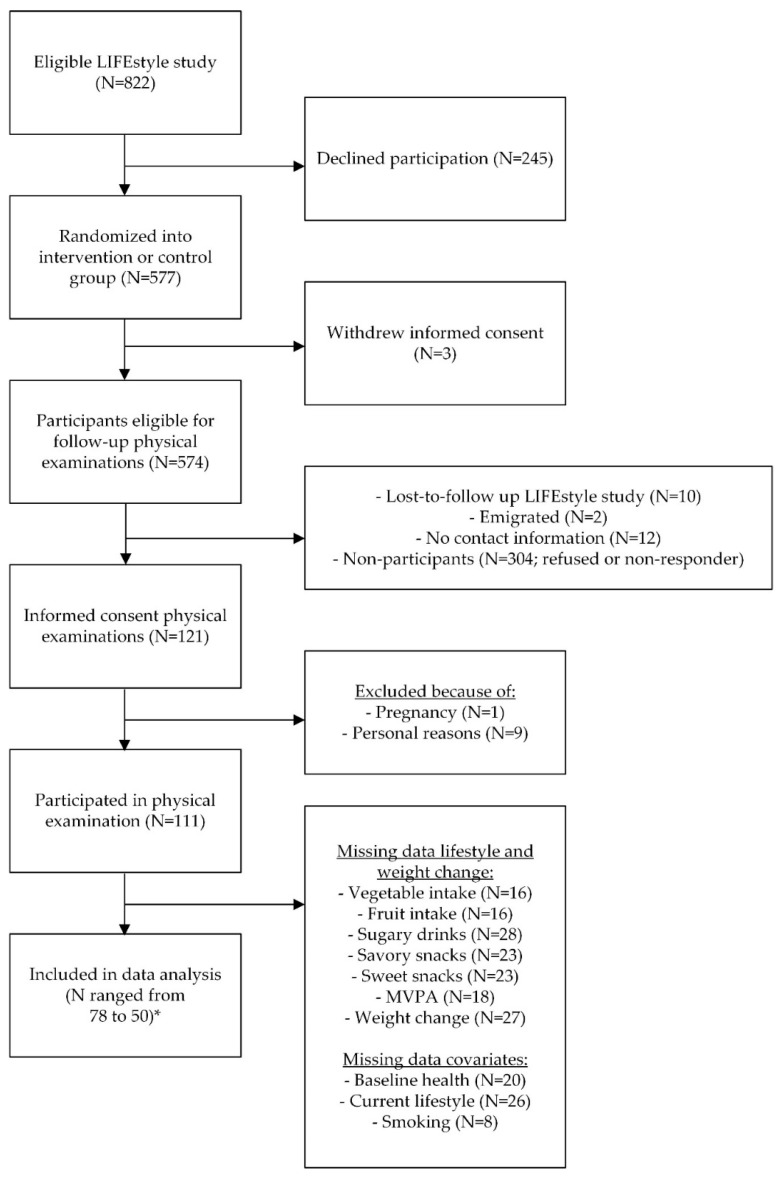
Flow-chart of the study participants. MVPA = Moderate to Vigorous Physical Activity * *N* depends on missing data in the combined models including independent variables and covariates.

**Table 1 nutrients-10-01953-t001:** Characteristics and cardiometabolic health of the study population (*N* = 111).

Age at follow-up (years; mean; SD)	36.4 (4.3)
Caucasian (*N*; %)	105 (94.6)
Education level (*N*; %)	
No education or primary school (4–12 years)	1 (0.9)
Secondary education	25 (23.6)
Intermediate Vocational Education	52 (49.1)
Higher Vocational Education or University	28 (26.4)
Body Mass Index at randomization (kg/m^2^; mean; SD)	35.7 (3.0)
Current smoker at follow-up (yes; *N*; %)	16 (15.5)
PCOS (yes; *N*; %)	43 (38.7)
Nulliparous at follow-up (yes; *N*; %)	21 (20.8)
Familial predisposition cardiovascular diseases (yes; *N*; %)	92 (89.3)
Gestational diabetes (yes; *N*; %) *	18 (17.5)
(Pre-)eclampsia (yes; *N*; %) *	16 (15.5)
HELLP syndrome (yes; %; *N*) *	7 (6.8)
Randomization group (intervention group; *N*; %)	50 (45.0)
Cardiovascular outcomes at follow-up	
Body Mass Index 3–8 years after randomization (kg/m^2^; mean; SD)	35.5 (5.3)
Systolic blood pressure (mmHg; mean; SD)	120.4 (14.4)
Diastolic blood pressure (mmHg; mean; SD)	81.7 (9.5)
Fat mass (% of total body weight; mean; SD)	43.1 (4.3)
Fat free mass (kg; mean; SD)	56.9 (6.5)
Pulse Wave Velocity (m/s; mean; SD)	7.2 (2.0)
Metabolic outcomes at follow-up	
Glucose (mmol/L; mean; SD)	5.3 (0.7)
Insulin (pmol/L; mean; SD)	81.7 (53.1)
HOMA-IR (mean; SD)	3.4 (2.8)
Triglycerides (mmol/L; mean; SD)	1.2 (0.8)
Total cholesterol (mmol/L; mean; SD)	4.6 (0.9)
LDL-C (mmol/L; mean; SD)	2.8 (0.8)
HDL-C (mmol/L; mean; SD)	1.3 (0.3)
Metabolic syndrome at follow-up (yes; *N*; %)	40 (40.0)

PCOS = Polycystic Ovary Syndrome; HELLP = syndrome characterized by hemolysis (H), elevated liver enzymes (EL) and low platelet count (LP); HOMA-IR = Homeostatic Model Assessment of Insulin Resistance; LDL-C = low-density lipoproteins cholesterol; HDL-C = high-density lipoproteins cholesterol. * Diagnosed with these pregnancy complications during any pregnancy in the past.

**Table 2 nutrients-10-01953-t002:** Change in dietary intake, physical activity and weight during the first six months of the LIFEstyle study in the study population included for follow-up *.

Change in	*N*	Mean (SD)	Median (IQR)
Vegetable intake (grams/day)	95	1.92 (58.70)	3.57 (−28.57; 35.71)
Fruit intake (grams/day)	95	25.56 (75.61)	14.29 (0.00; 85.71)
Sugary drinks (glasses/day)	83	−0.43 (1.81)	0.00 (−0.84; 0.21)
Savory snacks (handful/week)	88	−1.67 (5.57)	0.00 (−4.18; 0.57)
Sweet snacks (portion/week)	88	−0.15 (5.63)	0.00 (−2.61; 1.62)
Total MVPA (hour/week)	93	1.05 (12.1)	0.83 (−3.25; 6.13)
Body weight (kilograms)	84	−2.62 (5.10)	−2.10 (−4.68; 0.68)

MVPA = Moderate to Vigorous Physical Activity. * For all variables the change was calculated as preferably 6 months or otherwise 3 months minus baseline. If a higher score on the change variable is favorable or not depends on the independent variable of interest: A higher change score for vegetable intake, fruit intake, and MVPA is healthier, while a higher change score for sugary drink intake, savory and sweet snack intake, and weight change (higher means weight gain instead of weight loss) is unhealthier.

**Table 3 nutrients-10-01953-t003:** Association between the change in dietary intake and physical activity during the preconception lifestyle intervention and cardiometabolic health 3–8 years after randomization †.

	*N*	Crude Model *		Adjusted Model **	
		β (95% C.I.)	*p*	β (95% C.I.)	*p*
**Change in Vegetable Intake (10 g/day)**					
Body Mass Index (kg/m^2^)	76	−0.02 (−0.23; 0.19)	0.85	−0.04 (−0.25; 0.17)	0.71
Systolic blood pressure (mmHg)	74	0.50 (−0.07; 1.07)	0.08	0.49 (−0.11; 1.09)	0.11
Diastolic blood pressure (mmHg)	74	0.31 (−0.05; 0.68)	0.09	0.32 (−0.06; 0.70)	0.10
Fat mass (% of total body weight)	76	−0.04 (−0.21; 0.13)	0.64	−0.04 (−0.21; 0.14)	0.68
Fat free mass (kg)	76	0.14 (−0.14; 0.41)	0.33	0.10 (−0.18; 0.38)	0.47
Pulse Wave Velocity (m/s)	60	0.02 (−0.09; 0.14)	0.66	−0.004 (−0.13; 0.12)	0.95
Glucose (mmol/L)	64	−0.01 (−0.05; 0.02)	0.46	−0.01 (−0.05; 0.02)	0.52
Insulin (pmol/L)	63	0.56 (−1.35; 2.47)	0.56	1.03 (−0.94; 3.01)	0.30
HOMA-IR	61	0.001 (−0.11; 0.11)	0.99	0.02 (−0.10; 0.14)	0.72
Triglycerides (mmol/L)	64	−0.02 (−0.04; 0.01)	0.16	−0.01 (−0.03; 0.01)	0.33
Total cholesterol (mmol/L)	64	0.01 (−0.02; 0.05)	0.42	0.02 (−0.01; 0.06)	0.24
LDL-C (mmol/L)	64	0.02 (−0.01; 0.05)	0.15	0.02 (−0.01; 0.05)	0.14
HDL-C (mmol/L)	64	−0.002 (−0.01; 0.01)	0.70	−0.001 (−0.01; 0.01)	0.92
**Change in fruit intake (10 g/day)**					
Body Mass Index (kg/m^2^)	78	0.08 (−0.10; 0.26)	0.37	0.13 (−0.07; 0.32)	0.20
Systolic blood pressure (mmHg)	76	−0.09 (−0.59; 0.41)	0.72	−0.31 (−0.87; 0.25)	0.28
Diastolic blood pressure (mmHg)	76	0.00 (−0.32; 0.33)	>0.99	−0.04 (−0.41; 0.32)	0.81
Fat mass (% of total body weight)	78	0.10 (−0.05; 0.24)	0.19	0.12 (−0.04; 0.28)	0.13
Fat free mass (kg)	78	0.20 (−0.04; 0.44)	0.09	0.25 (−0.01; 0.52)	0.06
Pulse Wave Velocity (m/s)	63	−0.05 (−0.16; 0.05)	0.30	−0.05 (−0.16; 0.07)	0.39
Glucose (mmol/L)	66	0.03 (−0.01; 0.06)	0.12	0.01 (−0.02; 0.05)	0.41
Insulin (pmol/L)	65	0.52 (−1.26; 2.30)	0.56	0.52 (−1.40; 2.44)	0.59
HOMA-IR	63	−0.01 (−0.11; 0.10)	0.89	−0.02 (−0.14; 0.09)	0.72
Triglycerides (mmol/L)	66	−0.004 (−0.03; 0.02)	0.67	0.00 (−0.02; 0.02)	0.97
Total cholesterol (mmol/L)	66	0.03 (−0.002; 0.06)	0.07	0.02 (−0.01; 0.05)	0.27
LDL-C (mmol/L)	66	0.02 (−0.003; 0.05)	0.08	0.02 (−0.01; 0.04)	0.26
HDL-C (mmol/L)	66	0.003 (−0.01; 0.01)	0.57	−0.002 (−0.01; 0.01)	0.73
**Change in sugary drink intake (glass/day)**					
Body Mass Index (kg/m^2^)	62	0.46 (−0.71; 1.62)	0.44	0.42 (−0.79; 1.62)	0.49
Systolic blood pressure (mmHg)	60	2.04 (−1.55; 5.63)	0.26	2.34 (−1.34; 6.02)	0.21
Diastolic blood pressure (mmHg)	60	1.39 (−0.96; 3.74)	0.24	1.55 (−0.91; 4.00)	0.21
Fat mass (% of total body weight)	62	0.56 (−0.38; 1.50)	0.24	0.54 (−0.46; 1.54)	0.28
Fat free mass (kg)	62	0.58 (−1.15; 2.31)	0.50	0.48 (−1.32; 2.28)	0.60
Pulse Wave Velocity (m/s)	50	0.19 (−0.37; 0.76)	0.50	0.16 (−0.44; 0.75)	0.60
Glucose (mmol/L)	54	−0.03 (−0.27; 0.22)	0.84	−0.04 (−0.29; 0.21)	0.76
Insulin (pmol/L)	53	2.01 (−8.17; 12.19)	0.69	3.70 (−7.04; 14.44)	0.49
HOMA-IR	51	−0.20 (−0.90; 0.51)	0.57	−0.02 (−0.77; 0.72)	0.95
Triglycerides (mmol/L)	54	0.02 (−0.13; 0.16)	0.84	0.03 (−0.13; 0.18)	0.74
Total cholesterol (mmol/L)	54	0.07 (−0.11; 0.26)	0.44	0.06 (−0.13; 0.25)	0.56
LDL-C (mmol/L)	54	0.12 (−0.04; 0.29)	0.13	0.10 (−0.06; 0.27)	0.21
HDL-C (mmol/L)	54	−0.06 (−0.13; 0.01)	0.07	−0.07 (−0.13; 0.003)	0.06
**Change in savory snack intake (handful/week)**					
Body Mass Index (kg/m^2^)	72	0.05 (−0.20; 0.29)	0.72	0.21 (−0.06; 0.47)	0.12
Systolic blood pressure (mmHg)	70	0.25 (−0.47; 0.97)	0.49	0.45 (−0.35; 1.26)	0.26
Diastolic blood pressure (mmHg)	70	0.18 (−0.29; 0.65)	0.44	0.38 (−0.14; 0.90)	0.15
Fat mass (% of total body weight)	72	0.003 (−0.20; 0.20)	0.98	0.09 (−0.13; 0.31)	0.42
Fat free mass (kg)	72	−0.02 (−0.39; 0.36)	0.94	0.21 (−0.19; 0.60)	0.30
Pulse Wave Velocity (m/s)	58	0.04 (−0.06; 0.15)	0.44	0.04 (−0.08; 0.16)	0.49
Glucose (mmol/L)	62	−0.03 (−0.08; 0.02)	0.25	−0.02 (−0.08; 0.04)	0.57
Insulin (pmol/L)	61	−1.57 (−4.08; 0.95)	0.22	0.04 (−2.81; 2.88)	0.98
HOMA-IR	59	−0.16 (−0.32; −0.001)	0.049	−0.09 (−0.28; 0.09)	0.33
Triglycerides (mmol/L)	62	−0.02 (−0.06; 0.01)	0.17	−0.02 (−0.06; 0.02)	0.29
Total cholesterol (mmol/L)	62	−0.01 (−0.06; 0.04)	0.65	−0.02 (−0.08; 0.03)	0.42
LDL-C (mmol/L)	62	−0.001 (−0.04; 0.04)	0.95	−0.01 (−0.06; 0.04)	0.74
HDL-C (mmol/L)	62	−0.002 (−0.02; 0.02)	0.81	−0.01 (−0.03; 0.01)	0.42
**Change in sweet snack intake (portion/week)**					
Body Mass Index (kg/m^2^)	72	−0.10 (−0.29; 0.09)	0.30	−0.04 (−0.26; 0.17)	0.69
Systolic blood pressure (mmHg)	70	0.34 (−0.23; 0.92)	0.23	0.30 (−0.38; 0.98)	0.38
Diastolic blood pressure (mmHg)	70	0.25 (−0.11; 0.62)	0.17	0.30 (−0.13; 0.74)	0.17
Fat mass (% of total body weight)	72	−0.09 (−0.24; 0.06)	0.24	−0.04 (−0.21; 0.13)	0.65
Fat free mass (kg)	72	−0.04 (−0.32; 0.24)	0.76	0.04 (−0.28; 0.36)	0.79
Pulse Wave Velocity (m/s)	58	0.05 (−0.05; 0.16)	0.29	0.04 (−0.07; 0.14)	0.47
Glucose (mmol/L)	62	−0.03 (−0.07; 0.001)	0.06	−0.002 (−0.04; 0.04)	0.90
Insulin (pmol/L)	61	−1.16 (−2.84; 0.53)	0.18	0.57 (−1.37; 2.52)	0.56
HOMA-IR	59	−0.16 (−0.26; −0.06)	0.003	0.01 (−0.09; 0.12)	0.84
Triglycerides (mmol/L)	62	−0.02 (−0.04; 0.01)	0.17	−0.02 (−0.05; 0.01)	0.12
Total cholesterol (mmol/L)	62	0.02 (−0.01; 0.06)	0.18	0.01 (−0.04; 0.05)	0.71
LDL-C (mmol/L)	62	0.02 (−0.01; 0.05)	0.12	0.02 (−0.02; 0.05)	0.41
HDL-C (mmol/L)	62	0.004 (−0.01; 0.02)	0.52	−0.002 (−0.02; 0.01)	0.82
**Change in total MVPA (hour/week)**					
Body Mass Index (kg/m^2^)	76	−0.03 (−0.14; 0.09)	0.64	0.05 (−0.07; 0.16)	0.43
Systolic blood pressure (mmHg)	74	0.14 (−0.19; 0.46)	0.41	0.11 (−0.24; 0.47)	0.53
Diastolic blood pressure (mmHg)	74	−0.10 (−0.31; 0.10)	0.32	−0.09 (−0.31; 0.14)	0.44
Fat mass (% of total body weight)	76	−0.01 (−0.11; 0.08)	0.76	0.03 (−0.07; 0.13)	0.51
Fat free mass (kg)	76	0.09 (−0.06; 0.24)	0.25	0.14 (−0.02; 0.30)	0.09
Pulse Wave Velocity (m/s)	61	0.02 (−0.05; 0.09)	0.56	0.003 (−0.07; 0.07)	0.94
Glucose (mmol/L)	64	−0.01 (−0.03; 0.01)	0.26	−0.01 (−0.03; 0.02)	0.61
Insulin (pmol/L)	63	−0.59 (−1.74; 0.56)	0.31	−0.25 (−1.48; 0.99)	0.69
HOMA-IR	61	−0.03 (−0.10; 0.04)	0.39	−0.003 (−0.08; 0.07)	0.94
Triglycerides (mmol/L)	64	−0.001 (−0.02; 0.01)	0.86	0.002 (−0.01; 0.02)	0.76
Total cholesterol (mmol/L)	64	−0.01 (−0.03; 0.01)	0.38	−0.01 (−0.03; 0.02)	0.54
LDL-C (mmol/L)	64	−0.01 (−0.03; 0.01)	0.22	−0.01 (−0.03; 0.01)	0.42
HDL-C (mmol/L)	64	0.004 (−0.003; 0.01)	0.28	0.002 (−0.01; 0.01)	0.60
**Change in body weight during the intervention (kilograms)**					
Body Mass Index (kg/m^2^)	66	−0.12 (−0.35; 0.11)	0.31	−0.15 (−0.36; 0.06)	0.17
Systolic blood pressure (mmHg)	65	−0.27 (−0.89; 0.35)	0.38	−0.32 (−0.95; 0.32)	0.32
Diastolic blood pressure (mmHg)	65	−0.19 (−0.61; 0.23)	0.38	−0.22 (−0.65; 0.21)	0.31
Fat mass (% of total body weight)	66	−0.02 (−0.20; 0.16)	0.79	−0.04 (−0.22; 0.14)	0.65
Fat free mass (kg)	66	−0.14 (−0.42; 0.14)	0.32	−0.18 (−0.44; 0.08)	0.17
Pulse Wave Velocity (m/s)	53	0.02 (−0.07; 0.11)	0.63	0.02 (−0.07; 0.10)	0.69
Glucose (mmol/L)	55	0.001 (−0.05; 0.05)	0.98	−0.004 (−0.05; 0.04)	0.88
Insulin (pmol/L)	54	1.97 (−0.67; 4.61)	0.14	1.53 (−0.94; 3.99)	0.22
HOMA-IR	53	0.12 (−0.04; 0.28)	0.13	0.09 (−0.05; 0.24)	0.21
Triglycerides (mmol/L)	55	−0.01 (−0.03; 0.02)	0.65	−0.01 (−0.03; 0.02)	0.58
Total cholesterol (mmol/L)	55	−0.01 (−0.05; 0.04)	0.74	−0.004 (−0.05; 0.04)	0.87
LDL-C (mmol/L)	55	−0.01 (−0.05; 0.03)	0.61	−0.01 (−0.05; 0.03)	0.61
HDL-C (mmol/L)	55	0.01 (−0.01; 0.03)	0.24	0.01 (−0.001; 0.03)	0.06

MVPA = Moderate to Vigorous Physical Activity; HOMA-IR = Homeostatic Model Assessment of Insulin Resistance; LDL-C = low-density lipoproteins cholesterol; HDL-C = high-density lipoproteins cholesterol. † Change was calculated as preferably 6 months or otherwise 3 months minus baseline for vegetable intake, fruit intake and MVPA, and for sugary drink intake, snack intake and body weight, change was calculated as baseline minus preferably 6 months or otherwise 3 months. * The crude model is adjusted for baseline health outcomes and baseline lifestyle behavior. ** Model further adjusted for smoking (yes/no), current diet behavior depending on the variable of interest (e.g., in case of change in vegetable intake the model is adjusted for current vegetable intake, in case of MVPA the current dietary behavior is defined as current kcal intake) and current MVPA (min/day).

**Table 4 nutrients-10-01953-t004:** Association between the change in dietary intake and physical activity during the preconception lifestyle intervention and metabolic syndrome 3–8 years after randomization †.

Change in	*N*	Crude Model *		Adjusted Model **	
		OR (95% C.I.)	*p*-Value	OR (95% C.I.)	*p*-Value
Vegetable intake (10 g/day)	59	1.01 (0.90; 1.14)	0.87	1.01 (0.89; 1.14)	0.93
Fruit intake (10 g/day)	61	1.00 (0.90; 1.11)	0.99	1.01 (0.91; 1.12)	0.87
Sugary drink intake (glasses/day)	49	0.97 (0.38; 2.47)	0.94	0.75 (0.20; 2.87)	0.68
Savory snack intake (handful/week)	57	1.03 (0.89; 1.20)	0.67	1.17 (0.94; 1.44)	0.16
Sweet snack intake (portion/week)	57	1.01 (0.90; 1.12)	0.91	1.02 (0.88; 1.18)	0.76
Total MVPA (30 min/week)	59	1.00 (0.93; 1.07)	0.91	1.04 (0.95; 1.13)	0.42
Weight change (kg)	52	1.06 (0.90; 1.25)	0.49	0.98 (0.76; 1.27)	0.90

MVPA = Moderate to Vigorous Physical Activity. † Change was calculated as preferably 6 months or otherwise 3 months minus baseline for vegetable intake, fruit intake and MVPA, and for sugary drink intake, snack intake and body weight, change was calculated as baseline minus preferably 6 months or otherwise 3 months. * The crude model is adjusted for baseline metabolic syndrome (yes/no) and baseline lifestyle behavior. ** Model further adjusted for smoking (yes/no), current diet behavior depending on the variable of interest (e.g., in case of change in vegetable intake the model is adjusted for current vegetable intake, in case of MVPA the current dietary behavior is defined as current kcal intake) and current MVPA (min/day).

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
