# Peer review of "Cardiometabolic Health in Relation to Lifestyle and Body Weight Changes 3–8 Years Earlier"

_nutrients, 2018, doi:10.3390/nu10121953_

Reviewer 1 Report

In this manuscript, Elten et al reported the relationship between dietary/lifestyle intervention to future cadiometabolic health in obese women. The researchers found that though there is association between certain indexes (e.g. HOMA-IR) with the dietary intervention, the association was gone after correcting the current lifestyle. The methods are adequately described and limitations of this study are well discussed. Please see comments below.

1.     The title should convey a clear message. The current version "Changes in dietary intake, physical activity and body weight in obese infertile women and their cardiometabolic health 3-8 years later" is a collection of measurements- too descriptive without communicating the message well. Suggested title would be "6-month dietary intervention in obese infertile women doesn't affect cardiometabolic health 3-8 years later" or "Cardiometabolic health in relation to lifestyle intervention...". 

2.     Sample size: the hypothesis that limited sample size might change the conclusion is valid. The authors should put the effective sample size (i.e. 50-78) upfront (e.g. in the last paragraph of abstract) so that the readers would know what kind of sample size they are looking at. The current version mentioned total size (577) early but holds the real number till page 4. 

3.     (maybe I miss this) Can the author provide a table stating (if any) significant changes in current lifestyle triggered by the invention? Compare to the control group, did participants in the intervention group have better dietary control/ exercise for a longer time? Any quantification would be helpful. This might be an interesting topic for readers.

Author Response

Reviewer 1:

In this manuscript, Elten et al reported the relationship between dietary/lifestyle intervention to future cadiometabolic health in obese women. The researchers found that though there is association between certain indexes (e.g. HOMA-IR) with the dietary intervention, the association was gone after correcting the current lifestyle. The methods are adequately described and limitations of this study are well discussed. Please see comments below.

1.     The title should convey a clear message. The current version "Changes in dietary intake, physical activity and body weight in obese infertile women and their cardiometabolic health 3-8 years later" is a collection of measurements- too descriptive without communicating the message well. Suggested title would be "6-month dietary intervention in obese infertile women doesn't affect cardiometabolic health 3-8 years later" or "Cardiometabolic health in relation to lifestyle intervention...".

Response: Thank you for reviewing our manuscript. We changed the title of our manuscript in line with your second suggestion: Cardiometabolic health in relation to lifestyle and body weight changes 3-8 years earlier (see lines 2-6 of our manuscript including track changes).

2.     Sample size: the hypothesis that limited sample size might change the conclusion is valid. The authors should put the effective sample size (i.e. 50-78) upfront (e.g. in the last paragraph of abstract) so that the readers would know what kind of sample size they are looking at. The current version mentioned total size (577) early but holds the real number till page 4.

Respnse: We added the effective sample size into our abstract, lines 30-32:

“We examined if changes in dietary intake, physical activity and weight of obese infertile women during the first six months of the LIFEstyle trial were associated with cardiometabolic health 3-8 years later (N=50-78).”

3.     (maybe I miss this) Can the author provide a table stating (if any) significant changes in current lifestyle triggered by the invention? Compare to the control group, did participants in the intervention group have better dietary control/ exercise for a longer time? Any quantification would be helpful. This might be an interesting topic for readers.

Response: It is indeed interesting for readers to get insight in the changes in lifestyle triggered by the intervention. We studied the short term (up until 12 months after randomization) and long term (5.5 years after randomization) intervention effects on dietary intake and physical activity previously. We refer to these studies and the significant changes in lifestyle in our introduction section, lines 69-73:

“We previously showed that the intervention lowered the intake of high caloric snacks and beverages and increased physical activity in the short term [1], reduced body weight [2] and improved cardiometabolic health at the end of the six-month intervention period [3]. Furthermore, women allocated to the intervention group reported a lower energy intake at 5.5 years after randomization [4].”

References

1. van Elten, T. M.; Karsten, M. D. A.; Geelen, A.; van Oers, A. M.; van Poppel, M. N. M.; Groen, H.; Gemke, R. J. B. J.; Mol, B. W.; Mutsaerts, M. A. Q.; Roseboom, T. J.; Hoek, A.; group,  on behalf of the Life. study Effects of a preconception lifestyle intervention in obese infertile women on diet and physical activity; A secondary analysis of a randomized controlled trial. PLoS One 2018, 13, e0206888, doi:10.1371/journal.pone.0206888.

2. Mutsaerts, M. A. Q.; van Oers, A. M.; Groen, H.; Burggraaff, J. M.; Kuchenbecker, W. K. H.; Perquin, D. A. M.; Koks, C. A. M.; van Golde, R.; Kaaijk, E. M.; Schierbeek, J. M.; Oosterhuis, G. J. E.; Broekmans, F. J.; Bemelmans, W. J. E.; Lambalk, C. B.; Verberg, M. F. G.; van der Veen, F.; Klijn, N. F.; Mercelina, P. E. A. M.; van Kasteren, Y. M.; Nap, A. W.; Brinkhuis, E. A.; Vogel, N. E. A.; Mulder, R. J. A. B.; Gondrie, E. T. C. M.; de Bruin, J. P.; Sikkema, J. M.; de Greef, M. H. G.; ter Bogt, N. C. W.; Land, J. A.; Mol, B. W. J.; Hoek, A. Randomized Trial of a Lifestyle Program in Obese Infertile Women. N. Engl. J. Med. 2016, 374, 1942–1953, doi:10.1056/NEJMoa1505297.

3. van Dammen, L.*; Wekker, V.*; van Oers, A. M.; Mutsaerts, M. A. Q.; Painter, R. C.; Zwinderman, A. H.; Groen, H.; van de Beek, C.; Muller Kobold, A. C.; Kuchenbecker, W. K. H.; van Golde, R.; Oosterhuis, G. J. E.; Vogel, N. E. A.; Mol, B. W. J.; Roseboom, T. J.; Hoek, A.; LIFEstyle study group Effect of a lifestyle intervention in obese infertile women on cardiometabolic health and quality of life: A randomized controlled trial. PLoS One 2018, 13, e0190662, doi:10.1371/journal.pone.0190662.

4. van Elten, T. M.*; Karsten, M. D. A.*; Geelen, A.; Gemke, R. J. B. J.; Groen, H.; Hoek, A.; van Poppel, M. N. M.; Roseboom, T. J. Preconception lifestyle intervention reduces long term energy intake in women with obesity and infertility: A randomised controlled trial. Int. J. Behav. Nutr. Phys. Act. Under Rev. 2018.

Reviewer 2 Report

The manuscript presents data on lifestyle intervention outcomes 3-8 years post intervention with the outcome that current lifestyle reduced any association with cardiometabolic markers.  There is nothing novel in this manuscript and there are a large number of issues with the presentation.

 1) In the abstract and the results you state that you "adjusted for current lifestyle" but do not tell how this was accomplished.

2) It is inappropriate to say that the data is 'borderline significant'.   It is either significant or it is not.  You should just state the p values and let the reader make the assessment on their own.

3) Why is there such a large range in the time of follow-up assessment: 3-8 years? This makes the results difficult to compare, particularly with the small sample size.

4) You do not indicate how many of the women in each group gave birth after the initial intervention  trial.

5) It is very difficult to follow the sample size details.  You do not present figure 1 until the result section but in the methods do not provide any information which it makes it very difficult to understand the study.

6) Lines 94-115 read as though this is methodology from the original study.  I don't quite understand what this is.

7) Section 2.3 is too brief and run together.  Provide detail of the assessment, list the equipment used (devise name, manufacturer, where it is from). Line 143 - remove HOMA IR since it is described on line 144.

8) Line 200 - what do you mean by 'non-participants included'.  How can you include data of people who did not agree to participate?

9) All tables and figures should be left justified to make them easier to read.

10)  It seems to me that you are tryig to make this data appear to be significant when your outcome is null.  You need to carefully rework this manuscript to present the data clearly

Author Response

Reviewer 2:

The manuscript presents data on lifestyle intervention outcomes 3-8 years post intervention with the outcome that current lifestyle reduced any association with cardiometabolic markers. There is nothing novel in this manuscript and there are a large number of issues with the presentation.

1.     In the abstract and the results you state that you "adjusted for current lifestyle" but do not tell how this was accomplished.

Response: Thank you for your comments and for reviewing our manuscript. We added to our manuscript how we examined current lifestyle, see lines 222-230 of our manuscript including track changes:

“…… 1] smoking at follow-up (yes/no; self-reported by means of a questionnaire); 2] current dietary intake, depending on which food group is added into the model; e.g. if we examined the association between the change in vegetable intake during the first six months after randomization and current BMI, we corrected for current vegetable intake, and when examining the effect of MVPA we corrected for total current energy intake (kcal/day). Current dietary intake, i.e. dietary intake during follow-up, was assessed with the same 33-item FFQ as described previously [1]; 3] current MVPA (min/day), i.e. MVPA during follow-up, which was measured for seven consecutive days using the triaxial Actigraph wGT3X-BT or GT3X+ accelerometer [2]. Freedson cut-off points were used to determine the number of minutes per day in MVPA (≥1952 counts/min) [3].”

2.     It is inappropriate to say that the data is 'borderline significant'. It is either significant or it is not. You should just state the p values and let the reader make the assessment on their own.

Response: We deleted the sentence about borderline significance from our abstract, lines 39-41. Furthermore, we deleted the sentences about borderline significance in our results section, lines 274-280 & 288-296, in the first paragraph of our discussion section, lines 329-333, and in the last paragraph of our discussion section, lines 435-437.

Additionally, we changed our discussion section and do no longer mention “borderline significant”. We added the P-values to this section, see lines 353-361 & lines 368-373:

“There were multiple associations pointing towards our hypothesis that healthy changes in lifestyle are associated with more favorable cardiovascular health. Our findings that women with a higher intake of fruit (P=0.06) and more MVPA (P=0.09) had a higher fat free mass are in accordance with physiological mechanisms. Muscular activity stimulates the development and maintenance of lean muscle mass [4] and the high fiber content of the diet, associated with fruit intake, is related to lower body fat [5]. Furthermore, women who lost weight during the first six months after randomization tended to have higher HDL-cholesterol (P=0.06). This is in line with findings in other studies and may relate to links between lower visceral fat mass and higher HDL-cholesterol [6].”

“However, we also observed that women who decreased their sugary drink intake tended to have a lower HDL-cholesterol (P=0.06), which is not in line with our hypothesis and unexpected, assuming that a lower intake of sugary drinks causes a lower BMI, which is associated with improved HDL-cholesterol [6]. We do not know why we observed this: additional corrections for current BMI did not weaken this association and high sugary drink intake was not correlated with high MVPA levels, which is associated with higher HDL-cholesterol [7].”

3.     Why is there such a large range in the time of follow-up assessment: 3-8 years? This makes the results difficult to compare, particularly with the small sample size.

Response: There is indeed is a large variation in timing of follow-up measurements. This is because women were randomized into the preconception lifestyle intervention study between 2009 and 2012 and the follow-up study was conducted between 2015 and 2017. We agree that this makes results difficult to interpret. We therefore added the large range in time of follow-up as a limitation to our discussion section. To check if the large range in follow-up time affected our effect estimates and conclusion we additionally corrected our models for time between randomization and follow-up. This was, however, not the case. Also this information was added to the discussion section, see lines 418-422:

“Furthermore, there was a wide range (3-8 years) in the time between inclusion in the preconception lifestyle intervention study and our follow-up assessment of cardiometabolic health. This wide range might have affected the associations between lifestyle and body weight change with cardiometabolic health at follow-up. However, adding time in between randomization and follow-up into our models hardly changed the effect estimates.”

We added information regarding this sensitivity analysis to the method section, statistical analysis, lines 230-232:

“We additionally added …… and time between randomization and follow-up (years) into the adjusted model to see if this affected the effect estimates.”

Furthermore, we added this information to our result section, lines 310-311:

“Adding …… and time between randomization and follow-up (years) did not change effect estimates”

4.     You do not indicate how many of the women in each group gave birth after the initial intervention trial.

Response: During the initial intervention trial 76 out of 280 women (27.1%) in the intervention group and 100 out of 284 women (35.2%) in the control group gave birth to a healthy singleton at ≥37 weeks of gestation within 24 months after randomization.

5.     It is very difficult to follow the sample size details. You do not present figure 1 until the result section but in the methods do not provide any information which it makes it very difficult to understand the study.

Response: To give more insight in sample size details we moved figure 1 from the result section to the method section. We additionally present more sample size details in the method section accompanying figure 1, see lines 100-104:

“At 3-8 years after randomization, 574 women were eligible to participate in the follow‑up study of the LIFEstyle RCT [8]. During the follow-up study, data were collected using questionnaires and physical examinations. In the current study, we included all women that participated in the physical examinations (N=111; Figure 1).”

6.     Lines 94-115 read as though this is methodology from the original study. I don't quite understand what this is.

Response: Lines 94-115 explained which methods and instruments were used to measure dietary intake, physical activity and body weight during the first six months after randomization in the preconception lifestyle study. These measures were used to calculate the change in dietary intake, physical activity and body weight, which was studied in association to the cardiometabolic health 3-8 years after randomization. We changed the paragraph title and added additional information at the beginning of this paragraph to clarify why this information is included in our manuscript, see lines 113-116:

“2.2 Assessment of dietary intake, physical activity and body weight to calculate change

In order to calculate change in lifestyle and body weight over the first six months of the study, we used dietary intake, physical activity and body weight measures collected at baseline, 3 and 6 months after randomization in the lifestyle intervention study.”

7.     Section 2.3 is too brief and run together. Provide detail of the assessment, list the equipment used (devise name, manufacturer, where it is from). Line 143 - remove HOMA IR since it is described on line 144.

Response: We provided additional details of the assessment and equipment used to measure cardiometabolic health at follow-up in section 2.3, see lines 145-194:

“2.3 Cardiometabolic health at follow-up

Cardiometabolic health 3-8 years after randomization was examined in a mobile research vehicle by two researchers, using a standardized research protocol. Women were asked not to eat or drink from 90 minutes onwards before the mobile research vehicle arrived at their home. They were additionally asked not to drink caffeine containing beverages or to smoke from 12 hours onwards before the physical examination. Height and current weight were measured to calculate current BMI. Height was measured to the nearest 2.5px using a wall stadiometer (SECA 206; SECA, Germany) on bare feet, with heels flat on the ground at an angle of 90 degrees, head in Frankfort horizontal position and heels, back and shoulders straight against the wall. Current weight was measured in underwear to the nearest 0.1kg using a digital weighting scale (SECA 877; SECA, Germany), while the participant was standing still and looking straight ahead. All measurements were done twice, and in case of >12.5px difference in height and >0.5kg difference in weight, a third measurement was performed. After sitting quietly for 5 minutes, blood pressure was measured three times at heart level, at the non-dominant arm, using an automatic measurement device (Omron HBP-1300; OMRON Healthcare, The Netherlands) with appropriate cuff size. The three measurements were done using a 30 seconds time interval and women were not allowed to talk in between, move, cross their legs or tense their arm muscles. Body composition was measured twice by bio-electrical impedance (BIA; Bodystat 1500; Bodystat Ltd, Isle of Man, UK) after lying quietly for 5 minutes. On beforehand participants were asked to take of any jewelry, belts, piercings, etc. which could affect the BIA measurements. After cleaning the skin with alcohol, electrode strips were attached at the dorsal side of the left hand and foot with at least 3-125px in between the two electrodes. Women were instructed not to talk in between the measurements and attention was paid that arms and legs did not touch other parts of the body. A third measurement was performed in case the impedance or resistance differed >5Ω. Fat mass and fat free mass were calculated using equation of Kyle and colleagues [9]. Immediately after the BIA measurement, still in supine position, carotid-femoral pulse wave velocity (PWV) was measured twice using the Complior Analyse (Complior; Alam medical, France). Mechanotransducer censors were placed at the carotid artery on the right side and on the femoral artery on the left side. Blood pressure in lying position was measured once before the actual measurement started and entered in the Complior software. Directly after the measurement, distance between both censors was measured and also entered in the Complior software. In case of >10% difference in PWV between both measurements, a third measurement was done. The following equation was used to calculate PWV: PWV=0.8 x (distance between the carotis and fermoralis measuring site / Δ time between upstrokes of pressure [10].

Apart from the physical examinations, a trained nurse visited the participants at home to draw a venous blood sample after an overnight fast. All venous blood samples were analyzed at the biochemical laboratory of the Amsterdam UMC. We examined metabolic health by fasting serum concentrations of glucose (Roche cobas 8000, c702; Roche Diagnostics, Switzerland) and insulin (Centaur XP; Siemens, Germany), triglycerides (Roche cobas 8000, c702; Roche Diagnostics; Switzerland), total cholesterol (Roche cobas 8000, c502; Roche Diagnostics, Switzerland) and high density (HDL-C) lipoprotein cholesterol (Roche cobas 8000, c702; Roche Diagnostics, Switzerland). Low density (LDL-C) lipoprotein cholesterol was calculated using the following formula; [total cholesterol] – [high density lipoprotein cholesterol] 0.45 x [triglycerides]. Furthermore, insulin resistance (HOMA-IR) was calculated as fasting insulin concentration in μU/mL multiplied by fasting glucose concentration in mmol/L divided by 22.5. We additionally examined if metabolic syndrome was present or not. If present, participant met at least three of the following criteria determined by the American Heart Association: glucose ≥5.6 mmol/L, HDL-C<1.3 mmol/L, triglycerides ≥1.7 mmol/L, waist circumference ≥88 cm or blood pressure ≥130/85 mmHg [11].”

Furthermore, we removed HOMA-IR from line 143, see lines 183-187 in our manuscript including track changes.

8.     Line 200 - what do you mean by 'non-participants included'. How can you include data of people who did not agree to participate?

Response: We mean non-participants in our follow-up study, but participants in the initial study: the lifestyle intervention trial. We agree that this is not clear from this sentence and therefore clarified this, see lines  248-251:

“Baseline characteristics, collected during the LIFEstyle RCT, did not differ between participants (N=111) and non-participants (N=463) of the follow-up study, with exception of ethnicity and the change in sweet snack intake (Table S1).”

9.     All tables and figures should be left justified to make them easier to read.

Response: We did not change the typographical alignment of the tables and figures included in our manuscript, since the template of Nutrients specifically instruct authors to center all text in tables and figures (see Microsoft Word template at https://www.mdpi.com/journal/nutrients/instructions).

10.  It seems to me that you are tryig to make this data appear to be significant when your outcome is null. You need to carefully rework this manuscript to present the data clearly

Response: We have rewritten our abstract, result section and discussion section, also based on your previous comment about borderline significance. In short, in our abstract we deleted the sentences on borderline significance (lines 39-41), we present in our result section only the significant results (crude models HOMA-IR) and show that results became non-significant after adjustment for confounders (lines 270-296), we start our discussion section with our main finding and no longer concluded that there were borderline significant results in our discussion section (lines 325-333), replaced our paragraph in the discussion section about our null-finding as the first paragraph after the paragraph summarizing our results (lines 339-352) and we have rewritten the paragraphs in our discussion section regarding the borderline significant results (lines 353-361 and 368-379). Furthermore, we changed the conclusion of our manuscript, see lines 441-446 (conclusion section):

“To conclude, decreases in savory and sweet snack intake were associated with reduced insulin resistance 3-8 years later, but after adjustment for current lifestyle these associations disappeared. No other associations between changes in lifestyle or body weight during the first six months after randomization with cardiovascular health 3-8 years later were observed. Changing lifestyle is an important first step, but maintaining this change is needed to improve cardiometabolic health in the long-term.”

References

1. van den Brink, C.; Ocké, M.; Houben, A.; van Nierop, P.; Droomers, M.; RIVM rapport 260854008/2005 Validation of a Community Health Services food consumption questionnaire in the Netherlands. Available online: https://www.rivm.nl/bibliotheek/rapporten/260854008.pdf (accessed on Nov 4, 2018).

2. Santos-Lozano, A.; Santín-Medeiros, F.; Cardon, G.; Torres-Luque, G.; Bailón, R.; Bergmeir, C.; Ruiz, J.; Lucia, A.; Garatachea, N. Actigraph GT3X: Validation and Determination of Physical Activity Intensity Cut Points. Int. J. Sports Med. 2013, 34, 975–982, doi:10.1055/s-0033-1337945.

3. Freedson, P. S.; Melanson, E.; Sirard, J. Calibration of the Computer Science and Applications, Inc. accelerometer. Med. Sci. Sports Exerc. 1998, 30, 777–81.

4. Haskell, W. L.; Lee, I. M.; Pate, R. R.; Powell, K. E.; Blair, S. N.; Franklin, B. A.; Macera, C. A.; Heath, G. W.; Thompson, P. D.; Bauman, A. Physical Activity and Public Health: updated recommendation for adults from the American College of Sports Medicine and the American Heart Association. Med. Sci. Sport. Exerc. 2007, 39, 1423–1434, doi:10.1249/mss.0b013e3180616b27.

5. Slavin, J. L. Dietary fiber and body weight. Nutrition 2005, 21, 411–418, doi:10.1016/J.NUT.2004.08.018.

6. Rashid, S.; Genest, J. Effect of Obesity on High-density Lipoprotein Metabolism. Obesity 2007, 15, 2875–2888, doi:10.1038/oby.2007.342.

7. Loprinzi, P. D.; Addoh, O. The association of physical activity and cholesterol concentrations across different combinations of central adiposity and body mass index. Heal. Promot. Perspect. 2016, 6, 128–36, doi:10.15171/hpp.2016.21.

8.        van de Beek, C.; Hoek, A.; Painter, R. C.; Gemke, R. J. B. J.; van Poppel, M. N. M.; Geelen, A.; Groen, H.; Willem Mol, B.; Roseboom, T. J. Women, their Offspring and iMproving lifestyle for Better cardiovascular health of both (WOMB project): a protocol of the follow-up of a multicentre randomised controlled trial. BMJ Open 2018, 8, e016579, doi:10.1136/bmjopen-2017-016579.

9.        Kyle, U. G.; Genton, L.; Karsegard, L.; Slosman, D. O.; Pichard, C. Single prediction equation for bioelectrical impedance analysis in adults aged 20--94 years. Nutrition 2001, 17, 248–53.

10. Reference Values for Arterial Stiffness’ Collaboration Determinants of pulse wave velocity in healthy people and in the presence of cardiovascular risk factors: ‘establishing normal and reference values.’ Eur. Heart J. 2010, 31, 2338–2350, doi:10.1093/eurheartj/ehq165.

11. Grundy, S. M.; Cleeman, J. I.; Daniels, S. R.; Donato, K. A.; Eckel, R. H.; Franklin, B. A.; Gordon, D. J.; Krauss, R. M.; Savage, P. J.; Smith, S. C.; Spertus, J. A.; Costa, F. Diagnosis and management of the metabolic syndrome: an American Heart Association/National Heart, Lung, and Blood Institute scientific statement. Curr. Opin. Cardiol. 2006, 21, 1–6.

Reviewer 3 Report

The following are critiques that should be addressed:

1.      The rationale for assessing associations between changes in dietary intake, physical activity and weight during the first six months after randomization and cardiometabolic health 3-8 years later appears relatively weak. The length between 3-8 years is too broad. Authors may report the results by comparing after 3, 5, and/or 8 years intervention rather than the sum all of together.

2.      Per reviewer understanding, the initial 6 months intervention divided women into 2 groups (lifestyle intervention and control). The lifestyle intervention improved cardiometabolic health at the end of the six-month intervention period. In the present study, how many people were from lifestyle intervention group and how many people were from control group? Please justify the rationale to combine both if the sample size of each group was not similar. How would this affect the results?

3.      Page 2, lines 62-65: “Individual response to the lifestyle intervention varied largely among study participants. This variation enables us to assess associations between changes in dietary intake, physical activity and weight during the first six months after randomization and cardiometabolic health 3-8 years later.” This rationale is relatively weak. Please include rationales and related background in the introduction.

4.      Page 2, line 84: Please change “10.000” to “10,000”.

5.      Page 2, line 101: “Dietary intake was studied as the intake of 100 vegetables (raw as well as cooked; per 10 grams/day), fruits (per 10 grams/day)”. What are the justifications of using this cutoff (10 grams/day)? How about individuals who consumed less or more? Reviewer is trying to understand the FFQ questionnaire used in this study. However, the reviewer was unable to find the reference 12. In addition, the reference 16 is in non-English. Please include detailed information in the method section.

6.      Page 3, lines 100-103: Please describe detailed definition of all food categories. For example, sugar containing beverages (fruit juice and soda). There are different type of fruit juice, some are real juice from fruits and some are not. The present findings suggest that there was a borderline significant associations between decreases in sugary drink intake with lower HDL-cholesterol. Although composition of fruit juices is different from that of the edible portion of fruits, they contain polyphenols and vitamins from fruits. Many studies showed that fruit juices reduce cardiovascular risk factors, such as lowering blood pressure and improving blood lipid profiles. Reviewer strongly believe that beverage with added sugar or other sweetener including soda, fruit punch, lemonade and other “ades,” sweetened powdered drinks, and sports and energy drinks should fall in this category, but not the real fruit juice. Similar concern on savory snack (crisps, pretzels, nuts and peanuts). For example, several studies have shown that consumption of peanuts and nuts was associated with lower risk of total cardiovascular disease.

7.      Reviewer assumed that physical activity information was also collected from participant after 3-8 years intervention. However, reviewer could not find this information in the method section.

8.      Page 10, lines 251-252: Footnote “** Model further adjusted for smoking (yes/no), current diet behavior (and in case of MVPA for current kcal intake) and current MVPA (min/day).” It is not clear about the adjusted model used in statistical analysis. What is “current diet behavioral (and in case of MVPA for current kcal intake)”? Please describe/list all variables that were included in the model in the statistic section and footnote. 

9.      The discussion section needs to be expanded after addressing above comments from reviewer.

Author Response

Reviewer 3:

The following are critiques that should be addressed:

1.     The rationale for assessing associations between changes in dietary intake, physical activity and weight during the first six months after randomization and cardiometabolic health 3-8 years later appears relatively weak. The length between 3-8 years is too broad. Authors may report the results by comparing after 3, 5, and/or 8 years intervention rather than the sum all of together.

Response: Thank you for reviewing our manuscript. We expanded the introduction section on the rationale for assessing associations between changes in dietary intake, physical activity and weight and long term cardiometabolic health, see lines 63-67 in our manuscript including track changes:

“Despite the large variation in lifestyle and body weight change, the effects of a lifestyle intervention are often described as a randomized group comparison. By studying the effect of absolute change in lifestyle and body weight, more knowledge can be gathered on dose-response relationships and potential thresholds between lifestyle and body weight change with later life cardiometabolic health. This information will aid in improving primary prevention.”

We agree with the reviewer that we studied cardiometabolic health over a very broad time span (3-8 years). This is because women were randomized into the preconception lifestyle intervention study between 2009 and 2012 and the follow-up study was conducted between 2015 and 2017. We measured cardiometabolic health at follow-up once in every participant. Furthermore, given our small sample size we are, unfortunately, not able to report our results by comparing after 3, 5, and/or 8 years. We acknowledge that this is a limitation of our study and added this to our discussion section. We additionally performed a sensitivity analysis to check if our effect estimates changed if we added time of follow-up into our models, which was not the case. We added this information to our manuscript.

Discussion section, see lines 418-422:

“Furthermore, there is a wide range (3-8 years) in the time between inclusion in the preconception lifestyle intervention study and our follow-up assessment of cardiometabolic health. This wide range might have affected the associations between lifestyle and body weight change with cardiometabolic health at follow-up. However, adding time in between randomization and follow-up into our models hardly changed the effect estimates.”

We added information regarding the additional sensitivity analysis to the method section, statistical analysis, lines 230-232:

“We additionally added …… and time between randomization and follow-up (years) into the adjusted model to see if this affected the effect estimates.”

Furthermore, we added this information to our result section, lines 310-311:

“Adding …… and time between randomization and follow-up (years) did not change the effect estimates”

2.     Per reviewer understanding, the initial 6 months intervention divided women into 2 groups (lifestyle intervention and control). The lifestyle intervention improved cardiometabolic health at the end of the six-month intervention period. In the present study, how many people were from lifestyle intervention group and how many people were from control group? Please justify the rationale to combine both if the sample size of each group was not similar. How would this affect the results?

Response: Indeed, the initial study compared two groups of women and showed improved cardiometabolic health at the end of the six-month intervention period. Of the 111 women included in our study, 45% (N=50) were randomized to the intervention group and 55% (N=61) were randomized to the control group. We added this information to Table 1 (see the line “Randomization group (intervention group; N; %). Our data showed that also women in the control group changed their lifestyle and were successful in losing body weight during the trial (i.e. 10.5% of the women in the control group lost 5% of more of their original body weight over the first 6 months). We therefore combined both groups in our analysis. We added the rationale to combine both groups to the introduction section, lines 76-80:

“Also women allocated to the control group changed their lifestyle: 10.5% of the women in the control group lost 5% or more of their original body weight [1]. Hence we here investigate individual changes in lifestyle and body weight and relate these changes in dietary intake, physical activity and weight during the first six months after randomization to cardiometabolic health 3-8 years later.

Given the small sample size in our regression models (N=50-78) we are not able to stratify our models for randomization group. We agree that combining both groups might have affected our results and added this information to our discussion section, lines 343-346:

“Furthermore, it might be that there was not enough individual variation in the changes in lifestyle and body weight to find any associations with cardiovascular health at follow-up, which could be explained by the fact that women allocated to the control group participated to a larger extent in our follow-up study than women allocated to the intervention group.”

3.     Page 2, lines 62-65: “Individual response to the lifestyle intervention varied largely among study participants. This variation enables us to assess associations between changes in dietary intake, physical activity and weight during the first six months after randomization and cardiometabolic health 3-8 years later.” This rationale is relatively weak. Please include rationales and related background in the introduction.

Response: We included rationale and background to these lines in the introduction section, see lines 73-80:

“Individual responses to the lifestyle intervention varied largely among study participants. The mean weight change in the intervention group was -4.4kg and the standard deviation of 5.8kg underlines this large variation [1]. Also women allocated to the control group changed their lifestyle: 10.5% of the women in the control group lost 5% or more of their original body weight [1]. Hence we here investigate individual changes in lifestyle and body weight and relate these changes in dietary intake, physical activity and weight during the first six months after randomization to cardiometabolic health 3-8 years later.”

4.     Page 2, line 84: Please change “10.000” to “10,000”.

Response: We changed 10.000 into 10,000, see lines 98-99:

“Women were additionally advised to increase physical activity in daily life by taking at least 10,000 steps per day.”

5.     Page 2, line 101: “Dietary intake was studied as the intake of 100 vegetables (raw as well as cooked; per 10 grams/day), fruits (per 10 grams/day)”. What are the justifications of using this cutoff (10 grams/day)? How about individuals who consumed less or more? Reviewer is trying to understand the FFQ questionnaire used in this study. However, the reviewer was unable to find the reference 12. In addition, the reference 16 is in non-English. Please include detailed information in the method section.

Response: The intake of vegetables and fruits was measured in grams per day. In the results section, we show the effect on cardiometabolic health per 10 grams change of vegetable intake and fruit intake instead of per 1 gram (given that 1 gram is a very small amount and effect estimates were therefore difficult to interpret). It is confusing that we mention 10 grams in line 101 as vegetable and fruit intake is not measured per 10 grams. We therefore changed this line into (see lines 130-133):

“Dietary intake was studied as the intake of vegetables (raw as well as cooked; grams/day), fruits (grams/day), ……”

Furthermore, we now explain in our statistical analysis section that we studied the effect on cardiometabolic health per 10 grams increase of vegetable intake and fruit intake, see lines 200-202:

“We recalculated the intake of vegetables and fruits into 10 grams/day dividing the change by ten. Our regression models therefore display the effect on cardiometabolic health per 10 grams change of vegetable intake and fruit intake per day.”

Reference 12 referred to a published abstract. This paper has recently been accepted for publication by PloS One and is available online since November 7. We therefore changed this reference [2].

We changed reference 16 into a link to the complete report including an English abstract. There is no complete English version of this report. We therefore added relevant information from the report to our method section, lines 119-130:

“The first part was based on the standardized questionnaire on food consumption used for the Public Health Monitor in the Netherlands [3], asking about the type of cooking fats used, the consumed type of bread, frequency of breakfast use, frequency of consumption and portion size of vegetables, fruits and fruit juice. The questions on the intake of fruit, fruit juice and cooked vegetables were validated against two 24-hour recalls. The estimated intake of fruit and fruit juice consumption based on the questionnaire showed fairly strong comparability with the intake based on the two 24-hour recalls, however the comparability for cooked vegetables was weak [3]. The second part consisted of additional questions about savory and sweet snack intake and the intake of soda. For all foods, frequency of consumption per week or per months was asked and portion size was asked per standard household measure. The presented portion sizes and food groups in the current study were pre‑specified in the questions of the FFQ.”

6.     Page 3, lines 100-103: Please describe detailed definition of all food categories. For example, sugar containing beverages (fruit juice and soda). There are different type of fruit juice, some are real juice from fruits and some are not. The present findings suggest that there was a borderline significant associations between decreases in sugary drink intake with lower HDL-cholesterol. Although composition of fruit juices is different from that of the edible portion of fruits, they contain polyphenols and vitamins from fruits. Many studies showed that fruit juices reduce cardiovascular risk factors, such as lowering blood pressure and improving blood lipid profiles. Reviewer strongly believe that beverage with added sugar or other sweetener including soda, fruit punch, lemonade and other “ades,” sweetened powdered drinks, and sports and energy drinks should fall in this category, but not the real fruit juice. Similar concern on savory snack (crisps, pretzels, nuts and peanuts). For example, several studies have shown that consumption of peanuts and nuts was associated with lower risk of total cardiovascular disease.

Response: We understand and share your concerns. The definition of the food categories given in our manuscript are the same definitions as used in the 33-item FFQ. The FFQ used does not distinguish between type of fruit juice consumed (e.g. question on frequency = On how many days per week do you consume fruit juice [fresh juice and juice from a carton]?) or type of savory snack consumed (e.g. question on frequency = How often do you eat crisps, pretzels, nuts and peanuts?). We added this limitation to the discussion section, see lines 422-432:

“Finally, the 33-item FFQ pre-specified two food groups, fruit juice and savory snack intake, including foods known to have favorable as well as unfavorable effects on cardiometabolic health. The question on consumption of fruit juice does not distinguish between fresh fruit juice and sugar sweetened juice, while studies show that the consumption of fresh juice reduces cardiovascular risk factors due to, amongst others, the antioxidant effects and anti-inflammatory effects [4]. Furthermore, the question on savory snack consumption combines the intake of crisps, pretzels, nuts and peanuts into one question, while studies show that nuts and peanuts might be beneficial for cardiovascular health [5]. It might therefore be that an increase in these food groups represents a healthy change instead of an unhealthy change. We recommend future research to use a more extensive FFQ, not pre-specifying these foods into one food group.”

This might also explain why we observed a borderline significant associations between decreases in sugary drink intake with lower HDL-cholesterol. We added this information to our discussion section as well, lines 373-379:

“We were not able to analyze the effect of fresh fruit juice on HDL-cholesterol, because the 33-item FFQ do not ask specifically about fresh fruit juice. Evidence showed that fresh fruit juice might be associated with lower HDL-cholesterol [4]. It could therefore be that women specifically reduced their intake of sugar sweetened fruit juice, but not fresh fruit juice, which might have led to the unexpected association between decreases in sugary drinks and lower HDL-cholesterol.”

7.     Reviewer assumed that physical activity information was also collected from participant after 3-8 years intervention. However, reviewer could not find this information in the method section.

Response: Indeed, physical activity information was also collected from our participants at 3-8 years after randomization. We added this information to our method section, lines 227-230:

“…3] current MVPA (min/day), i.e. MVPA during follow-up, which was measured for seven consecutive days using the triaxial Actigraph wGT3X-BT or GT3X+ accelerometer [6]. Freedson cut-off points were used to determine the number of minutes per day in MVPA (≥1952 counts/min) [7].”

8.     Page 10, lines 251-252: Footnote “** Model further adjusted for smoking (yes/no), current diet behavior (and in case of MVPA for current kcal intake) and current MVPA (min/day).” It is not clear about the adjusted model used in statistical analysis. What is “current diet behavioral (and in case of MVPA for current kcal intake)”? Please describe/list all variables that were included in the model in the statistic section and footnote.

Response: We added additional information to our footnote to make clear what we mean with “current diet behavior” and “in case of MVPA for current kcal intake”, see lines 304-307:

“**Model further adjusted for smoking (yes/no), current diet behavior depending on the variable of interest (e.g. in case of change in vegetable intake the model is adjusted for current vegetable intake, in case of MVPA the current dietary behavior is defined as for current kcal intake) and current MVPA (min/day).”

In line with these changes, we also changed the footnote of table 4, see lines 319-323.

9.     The discussion section needs to be expanded after addressing above comments from reviewer.

Response: We expanded our discussion section addressing the above comments, see lines 343-346; lines 373-379; lines 418-422; lines 422-432.

References

1. Mutsaerts, M. A. Q.; van Oers, A. M.; Groen, H.; Burggraaff, J. M.; Kuchenbecker, W. K. H.; Perquin, D. A. M.; Koks, C. A. M.; van Golde, R.; Kaaijk, E. M.; Schierbeek, J. M.; Oosterhuis, G. J. E.; Broekmans, F. J.; Bemelmans, W. J. E.; Lambalk, C. B.; Verberg, M. F. G.; van der Veen, F.; Klijn, N. F.; Mercelina, P. E. A. M.; van Kasteren, Y. M.; Nap, A. W.; Brinkhuis, E. A.; Vogel, N. E. A.; Mulder, R. J. A. B.; Gondrie, E. T. C. M.; de Bruin, J. P.; Sikkema, J. M.; de Greef, M. H. G.; ter Bogt, N. C. W.; Land, J. A.; Mol, B. W. J.; Hoek, A. Randomized Trial of a Lifestyle Program in Obese Infertile Women. N. Engl. J. Med. 2016, 374, 1942–1953, doi:10.1056/NEJMoa1505297.

2. van Elten, T. M.; Karsten, M. D. A.; Geelen, A.; van Oers, A. M.; van Poppel, M. N. M.; Groen, H.; Gemke, R. J. B. J.; Mol, B. W.; Mutsaerts, M. A. Q.; Roseboom, T. J.; Hoek, A.; group,  on behalf of the Life. study Effects of a preconception lifestyle intervention in obese infertile women on diet and physical activity; A secondary analysis of a randomized controlled trial. PLoS One 2018, 13, e0206888, doi:10.1371/journal.pone.0206888.

3. van den Brink, C.; Ocké, M.; Houben, A.; van Nierop, P.; Droomers, M.; RIVM rapport 260854008/2005 Validation of a Community Health Services food consumption questionnaire in the Netherlands.

4. Zheng, J.; Zhou, Y.; Li, S.; Zhang, P.; Zhou, T.; Xu, D.-P.; Li, H.-B. Effects and Mechanisms of Fruit and Vegetable Juices on Cardiovascular Diseases. Int. J. Mol. Sci. 2017, 18, 555, doi:10.3390/ijms18030555.

5. de Souza, R. G. M.; Schincaglia, R. M.; Pimentel, G. D.; Mota, J. F. Nuts and Human Health Outcomes: A Systematic Review. Nutrients 2017, 9, 1311, doi:10.3390/nu9121311.

6. Santos-Lozano, A.; Santín-Medeiros, F.; Cardon, G.; Torres-Luque, G.; Bailón, R.; Bergmeir, C.; Ruiz, J.; Lucia, A.; Garatachea, N. Actigraph GT3X: Validation and Determination of Physical Activity Intensity Cut Points. Int. J. Sports Med. 2013, 34, 975–982, doi:10.1055/s-0033-1337945.

7. Freedson, P. S.; Melanson, E.; Sirard, J. Calibration of the Computer Science and Applications, Inc. accelerometer. Med. Sci. Sports Exerc. 1998, 30, 777–81.